# RNA-Sequencing, Physiological and RNAi Analyses Provide Insights into the Response Mechanism of the *ABC*-Mediated Resistance to *Verticillium dahliae* Infection in Cotton

**DOI:** 10.3390/genes10020110

**Published:** 2019-02-01

**Authors:** Qi Dong, Richard Odongo Magwanga, Xiaoyan Cai, Pu Lu, Joy Nyangasi Kirungu, Zhongli Zhou, Xingfen Wang, Xingxing Wang, Yanchao Xu, Yuqing Hou, Kunbo Wang, Renhai Peng, Zhiying Ma, Fang Liu

**Affiliations:** 1State Key Laboratory of Cotton Biology/Institute of Cotton Research, Chinese Academy of Agricultural Science (ICR, CAAS), Anyang, Henan, 455000, China; dongqi9208@163.com (Q.D.); magwangarichard@yahoo.com (R.O.M); caixy@cricaas.com.cn (X.C.); lupu1992@163.com (P.L.); joynk@cricaas.com.cn (J.N.K.); zhouzl@cricaas.com.cn (Z.Z.); wangxx@cricaas.com.cn (X.W.); xuyanchao2116@163.com (Y.X.); houyq@cricaas.com.cn (Y.H.); wkbcri@cricaas.com.cn (K.W.); 2College of Agronomy, Hebei Agricultural University/North China Key Laboratory for Crop Germplasm Resources of Ministry of Education, Baoding, Hebei, 071001, China; cotton@hebau.edu.cn; 3School of Biological and Physical Sciences (SBPS), Main Campus, Jaramogi Oginga Odinga University of Science and Technology (JOOUST), Main Campus, P.O. Bondo, Box 210-40601, Kenya; 4Biological and Food Engineering, Anyang Institute of technology, Anyang 455000, Henan, China; aydxprh@163.com

**Keywords:** cotton, *Verticillium* wilt, *Verticillium dahliae*, oxidant and antioxidant enzymes, immunity, virus induced gene silencing, ATP-binding cassette *(ABC)* proteins

## Abstract

*Verticillium* wilt that is caused by *Verticillium dahliae,* does result in massive annual yield losses and fiber quality decline in cotton. Control by conventional mechanisms is not possible due to a wide host range and the longevity of dormant fungi in the soil in the case of absence of a suitable host. Plants have developed various mechanisms to boost their immunity against various diseases, and one is through the induction of various genes. In this research, we carried out RNA sequencing and then identified the members of the adenosine triphosphate (ATP)-binding cassette (ABC) proteins to be critical in enhancing resistance to *V. dahliae* infection. A total of 166 proteins that are encoded by the *ABC* genes were identified in *Gossypium raimondii* with varying physiochemical properties. A novel *ABC* gene, *Gorai.007G244600* (ABCF5), was found to be highly upregulated, and its homolog in the tetraploid cotton *Gh_D11G3432* (ABCF5), was then silenced through virus induced gene silencing (VIGS) in *G. hirsutum*, tetraploid upland cotton. The mutant cotton seedlings ability to tolerate *V. dahliae* infection was significantly reduced. Based on the evaluation of oxidant enzymes, hydrogen peroxide (H_2_O_2_) and malondialdehyde (MDA) showed significantly increased levels in the leaves of the mutant compared to the wild type. In addition, antioxidant enzymes, peroxidase (POD), catalase (CAT), and superoxide dismutase (SOD) concentrations were reduced in the mutant cotton leaves after treatment with *V. dahliae* fungi as compared to the wild type. Moreover, expression levels of the biotic stress genes, cotton polyamine oxidase (*GhPAO*), cotton ribosomal protein L18 (*GhRPL18*), and cotton polygalacturonase-inhibiting protein-1 (*GhPGIP1*), were all downregulated in the mutant but they were highly upregulated in the various tissues of the wild cotton seedlings. This research has shown that *ABC* genes could play an important role in enhancing the immunity of cotton to *V. dahliae* infection, and thus can be explored in developing more resilient cotton genotypes with improved resistance to *V. dahliae* infection in cotton.

## 1. Introduction

*Verticillium* wilt is a serious vascular disease that is often compared to other viral diseases of both animals and human alike; it is caused by the soil-borne fungus known as *Verticillium dahliae* [1]. The virulent nature of the fungus is so high with a diverse host range of more than 400 plant species, including a number of economically important crops, such as cotton and other food crops [2]. The fungus that causes *V.* wilt in plants can survive for long periods of time in the soil as microsclerotia in the absence of a suitable host [3]. The fungus get its entry point into the plants through the roots, and it then spreads through the xylem vessels, upon infection, it undergoes rapid multiplication, and spread to shoots and leaves of the infected plant. The prolific growth of the fungus within the xylem vessels affects the plants ability to transport water and nutrients. [4]. In cotton, the disease infection is seen at relatively early stages of plant development, though the symptoms become more severe after flowering. On the cotton plants, the symptoms are necrotic lesion on the leaf, stunted growth, premature defoliation, discoloration of the vascular system, ball abortion, and even plant death [5].

Cotton is the number one textile indispensable source of raw materials [6]. Its growth and production has been hampered by fungus infection. It is estimated that the losses incurred due to disease infection is extremely high in major cotton growing regions, which is due to no remedy once the disease occurs [7]. Several attempts have been made to developed resistant cotton cultivars using resistant germplasm, such as *Gossypium barbadense,* also known as the Egyptian or pima cotton [8]. However, little progress has been achieved so far due to hybrid breakdown and sterility in the interspecific crosses [8]. Some farming practices, such as crop rotation, has not yielded much, due to the persistent nature of the microsclerotia in soil and the lack of an effective chemical means of combating the pathogens [4]. The only remedy to solve the problem of the disease is through the identification and utilization of disease-resistance genes [9,10,11,12]. Other proteins that are encoded by disease resistance genes, such as the myeloblastosis (MYB) transcription factors [13], mitogen-activated protein kinases [10], major latex proteins [14], WRKY transcription factors [15], polyamine oxidases, subtilase [16], receptor-like kinase [17], and BRI1-associated receptor kinase [18], have been found to have a profound effect on enhancing resistance to *V. dahliae* infection in cotton.

Being the fungus targets the transporting channel, several transporter proteins have been found to have a functional role in enhancing biotic and abiotic stress tolerance, for instance, adenosine triphosphate (ATP)-dependent binding cassette transporter G family member16 has been found to enhance resistance to virulent bacterial pathogen *Pseudomonas syringae* which blocks the functioning of the plant guard cell [19,20]. Moreover, the ATP-binding cassette (ABC) transporter superfamily is one of the largest protein families with wide distribution in both lower and higher organisms [21]. The ABCs are mainly responsible for the uptake of nutrients, such as amino acids, sugars, fatty acids, and essential metals in plants [22], while in humans, the ABC exporters mediate the export of lipids, vitamins, proteins, carbohydrates, and cholesterols [23]. The ABC transporter families are categorized into different sub groups that are based on their structural features, namely, the pleiotropic drug resistance (PDR), multidrug resistance (MDR), and multidrug resistance associated protein (MRP) [24]. These proteins have been found to have a functional role in plants in enhancing both the biotic and abiotic stress factors.

The ABC transporter has three structural types, including full-molecular transporters, semi-molecular proteins, and soluble transporters. The full-molecule ABC transporter has two nucleotide-binding domains (NBDs) and two transmembrane domains (TMDs), while the half-molecular transporter contains only one transmembrane domain and one nucleotide binding domain, the third structural type has only the nucleotide binding domain, but no transmembrane domain, it is also known as the soluble protein type [25]. Most ABC transporters bind to adenosine triphosphate (ATP) and they release energy by hydrolyzing ATP to regulate the transmembrane transport of the substance [26]. The functions of the *ABC* genes have been investigated in various plants such as rice [27], Arabidopsis [28], tobacco [29], and soybean [30]. Studies have confirmed that *ABC* transporters play important roles in plant pathogenic microbial responses [31], nutrient transport in plant-microbial symbiotic interactions [32], the regulation of heavy metals [33], and the transport of secondary metabolites [34], and even in the transport of plant hormones and exogenous chemicals [35]. Based on the broader roles of the plant ABCs, no information has been made available on their role in cotton in relation to *Verticillium* wilt. There are few reports on the identification and prediction of cotton *ABC* transporters. Therefore, understanding the composition of the cotton *ABC* transporter gene family and its expression in the pathogen-plant interaction process is particularly important to further reveal its transport of nutrients in plant immunity and its interaction with microorganisms. Based on our transcriptome data of wild cotton and the previous sequencing of *G. raimondii*, we selected three cotton diploid wild species with varying sensitivity to *Verticillium dahliae* infection. *Gossypium thurberi, Gossypium trilobum,* and *Gossypium raimondii*. *G. thurberi* has been found to have higher resistance ability to *V. dahliae* infection, while *G. trilobum* has a relatively mild resistance [36]. Being the two wild cotton species of the D genome has not been sequenced; *G. raimondii* was applied as the reference genome. Concretely, the *ABC* transporter in D genome wild cotton was identified and its function was studied. The outcome of this research not only lays the foundation for the study of the role of the *ABC* genes, but also provides a reference for future studies on the mechanism of the *ABC* genes in enhancing autoimmune to *Verticillium dahliae* in cotton.

## 2. Materials and Methods

### 2.1. Plant Materials, Growth Conditions and Innoculation with *V. dahliae*

Three accessions, D1-5, D5-lz, and D8-7, representing three wild diploid species in *Gossypium* of the D genome, *G. raimondii* (D_5_)*, G. trilobum* (D_8_), and *G. thurberi* (D_1_), respectively, were used in this study. Some of them are known to harbor some vital genes that enhance resistance to *V. dahliae*, while *G. raimondii* is known to be highly susceptible to *V. dahliae* infection [37]. The plant materials were obtained from the National Wild Cotton Nursery, which is located in Sanya City, Hainan Island, China, which is managed by the Institute of Cotton Research, Chinese Academy of Agricultural Sciences (ICR-CAAS). The cotton seeds were delinted by the use of sulfuric acid, sterilized by using 1% of sodium hypochlorite for 15 min, and then washed with sterilized water three times to ensure that the seeds were free of pathogens. To ensure maximum germination, a small slit was made on the seed coat, and they were then pre-germinated in the incubator using sterilized moist filter paper for 2–3 days until the radicle was approximately 1 cm long and the condition of the incubator was set at 28 °C with > 90% ambient humidity. The seedlings were then transplanted into sterilized vermiculite filled in small conical pots, with dimensions of 5 cm bottom region, 7 cm top region, and 8 cm in depth, and at one true leaf stage, the plants were inoculated with 8 mL suspension of *V. dahliae* at 1 × 10^7^ conidia per mL [38]. The infectious nature of the *V. dahliae* on the leaf, stem, and root tissues of the three cotton seedlings was monitored, the leaf and stem tissues were examined after 25 days of post inoculation (dpi) of the seedlings at one true leaf stage, while the root tissues were examined by applying green fluorescent protein (GFP). The samples were then taken at 0 h, 12 h, and 48 h for RNA sequencing and physiological parameters evaluation, while for quantitative real-time PCR (RT-qPCR) analysis, the samples were collected at 0 h, 12 h, and 48 h. The virulent *V. dahliae* strain T5 [39] and *V. dahliae* with green fluorescent protein Vd-GFP [17] were provided by the Cotton Genetics and Breeding Laboratory of Hebei Agricultural University, Henan province, China.

### 2.2. RNA Isolation, Library Construction and Sequencing

The leaf, root, and stem tissues of the three cotton species were sampled for RNA-seq analysis with a total of 81 samples. To simplify the description, we designated three replicates of the samples for RNA sequencing corded, as follows; for *G. raimondii*, root samples (Gr_0R, Gr_12R, Gr_48R), stem (Gr_0S, Gr_12S, Gr_48S), and the leaf (Gr_0L, Gr_12L, Gr_48L), *G. thurberi* root (Gth_0R, Gth_12R, Gth_48R), stem (Gth_0S, Gth_12S, Gth_48S), and the leaf (Gth_0L, Gth_12L, Gth_48L); and finally, for the *G. trilobum* root (Gtr_0R, Gtr_12R, Gtr_48R), stem (Gtr_0S, Gtr_12S, Gtr_48S), and the leaf (Gtr_0L, Gtr_12L, Gtr_48L). Total RNA for the samples was extracted using Trizol reagent (Invitrogen, Carlsbad, CA, USA), as per the manufacturer’s protocol. The total RNA quantity and purity were analysis of Bioanalyzer 2100 (Agilent, Santa Clara, CA, USA) and RNA 6000 Nano Lab Chip Kit (Agilent, Santa Clara, CA, USA) with RIN number >7.0. Approximately 10 µg of total RNA representing a specific adipose type was subjected to isolate Poly (A) mRNA with poly-T oligo attached magnetic beads (Invitrogen, Carlsbad, CA, USA). Following purification, the mRNA was fragmented into small pieces using divalent cations under elevated temperature. Subsequently, the cleaved RNA fragments were reverse-transcribed to create the final cDNA library in accordance with the protocol for the mRNA Seq sample preparation kit (Illumina, San Diego, CA, USA), and the average insert size of the paired-end libraries was 300 bp (±50 bp). We performed the paired-end sequencing on an Illumina Hiseq4000 at the (LC Sciences, Mundelein, IL, USA), following the vendor’s recommended protocol.

### 2.3. Oxidant and Antioxidant Enzyme Assays

From each of the three cotton species, *G. raimondii, G. thurberi,* and *G. trilobum,* fresh leaves were harvested at 0 h, 12 h, and 48 h of post inoculation. The fresh leaf samples of 0.5 g were ground and vortexed into fine powder in a mortar containing 1.5 mL of 50 mM sodium phosphate buffer solution with pH of 7.8. The buffer solution was a mixture of 2 mM Ethylenediaminetetraacetic acid (EDTA), 5 mM β-mercaptoethanol, and 4% (*w*/*v*) polyvinylpyrrolidone-40. The grounded leaf samples in a buffer solution was then transferred into 7 mL test tubes, and then shaken for the mixture to homogenized, and the homogenate was then centrifuged at 12,000 rotations per minute (rpm) for 20 min at 4 °C. After centrifuging, the upper layer (supernatant) was then pipetted and stored at −80 °C for the evaluation of oxidant and antioxidant enzyme assays. The activities of antioxidant enzymes, including peroxidase (POD), catalase (CAT), and superoxide dismutase (SOD), were measured as outlined by the guaiacol colorimetric methods, potassium permanganate titration, and NBT-illumination method, respectively [40]. In addition, two oxidant enzymes were evaluated, hydrogen peroxide (H_2_O_2_) and malondialdehyde (MDA), as outlined by Lu et al. [41]. The oxidant and antioxidant enzymes evaluation data were analyzed with ANOVA using Statistical Package for the Social Sciences (SPSS).

### 2.4. Quantitative Real-Time PCR (RT-qPCR) Analysis

Significant ABC differentially expressed genes (DEGs) were selected to synthesize cDNA using an M-MLV First Strand kit (Invitrogen) with three biological replicates. The *G. raimondii*, Gractin7 gene with the forward sequence “ATCCTCCGTCTAGACCTTG” and the reverse sequence of “TGTCCATCAGGCAACTCAT” was used for internal normalization. The primers were designed using the primer 5.0 software (Appendix A). RT-qPCR was performed in iQ5 system (Bio-Rad, Hercules, CA, USA) using a 20 µL reaction mixture, including 2 µL cDNA (1:10 dilution), 10 µL of 2× TransStart Top Green qPCR SuperMix (TransGen, Beijing, China), 1.0 µL of each primer (10 µmol/µL), and 6 µL ddH_2_O. Reaction conditions were carried out with 95 °C for 10 min, followed by 40 cycles of 95 °C for 5 s, 59 °C for 15 s, and 72 °C for 30 s. The relative gene expression levels were calculated while using the 2−ΔΔCt method [42]. The expression differences between the cotton species, *G. raimondii*, *G. trilobum*, and *G. thurberi* at 0 h, 12 h, and 48 h post inoculation were analyzed by the t test.

### 2.5. RNA-Seq Data Quality Assessment and Bioinformatics Analysis

Raw sequence data for the three cotton genotypes were processed through Perl scripts that were developed by Biomarker Technologies Co. Ltd. (Beijing, China) to churn out low quality reads, the low quality reads are those reads with more than 20% of bases having Q values of less or equal to 20 or those with unknown or ambiguous sequence contents, such as N and exceeding 5%, as previously outlined by Magwanga et al. [43]. *G. raimondii* was applied as the reference genome, the genome sequences were obtained from *Gossypium*. Genomics ftp website (http://ftp.bioinfo.wsu.edu/species/Gossypium_raimondii/JGI_221_G.raimondii_Dgenome/assembly/G.raimondii_JGI_221_v2.0.assembly.fasta.gz)). The high-quality reads were mapped to the reference genome using HISAT tool [44] with default settings. Subsequently, transcripts were built by String Tie software [45] to identify the known and novel transcripts under default parameters. The fragments per kilobase of the transcript sequence per million mapped sequenced method (FPKM) was used to determine the transcript expression levels in all of the samples and the correlations among replicates [46]. The R-language Ballgown package was used to analyze the differences between the assembled and quantified StringTie genes (log 2 fold change ≥1 with P value of less than 0.05) [47]. The categories, amount of transcription factors (TFs), and the heat map were predicted using the iTAK software [48]. The RNA sequencing data can be accessible through GEO Series accession number GSE123175 (https://www.ncbi.nlm.nih.gov/geo/query/acc.cgi?acc=GSE123175).

### 2.6. Functional Annotation and Enrichment Pathway Analysis of DEGs

The differentially expressed genes (DEGs) were further analyzed by carrying out gene ontology and KEGG enrichment (http://www.genome.jp/kegg/), this was done in order to functionally characterized the genes in relation to biological processes (BP), cellular component (CC), and molecular functions (MF), and or to determine the possible pathways or signal transduction pathways in which the genes could possibly be involved in. The DEGs were submitted to GO analysis of enrichment using the GO-seq R package, as per the Wallenius’ non-central hyper-geometric distribution [49], KOBAS software [50] was applied to validate the statistical enrichment of DEGs in the KEGG pathways. Finally, the MapMan software version X4 (http://MapMan.gabipd.org) was used to develop the graphical presentations of metabolism and regulatory pathways.

### 2.7. Identification of ABC Gene Family, Chromosomal Mapping and Subcellular Localization Prediction of the ABC Proteins in G. raimondii

The conserved domain of the ABC protein was downloaded using a hidden Markov model (HMM)(PF00005). To identify the ABC proteins in cotton, mainly in D genome, the HHM profile of the ABC protein was subsequently used as a query in an HMMER search (http://hmmer.org/) against the genome sequences of *G. raimondii*. *G. raimondii* and *Theobroma cacao* genomes were downloaded from Phytozome (https://www.phytozome.net/), *Arabidopsis thaliana* obtained from TAIR (http://www.arabidopsis.org/) and those of *Oryza sativa* were downloaded from the rice genome website (http://rice.plantbiology.msu.edu/) with an E-value <0.01. All of the redundant sequences were discarded from further analysis based on cluster W17 alignment results. Furthermore, SMART and PFAM databases were used to verify the presence of the *ABC* gene domains. The isoelectric points and molecular weights of the ABC proteins were estimated with the ExPASy Server tool (http://web.expasy.org/). The chromosomal distribution of the *ABC* genes was mapped on the cotton chromosomes based on gene position by mapchart 2.2 software [51]. In addition, subcellular location prediction of GrABC proteins was determined with online tools TargetP1.1 server (http://www.cbs.dtu.dk/services/TargetP/) and Protein Prowler Subcellular localization Predictor version 1.2 (http://bioinf.scmb.uq.edu.au:8080/pprowler_webapp_1-2/) validations and the determination of the possible cell compartmentalization, as obtained by the two software programs, was done by WolF PSORT (https://wolfpsort.hgc.jp).

### 2.8. Phylogenetic Analyses and Gene Structure Organization of the ABC Proteins in Cotton

The full protein sequences of the ABC for diploid cotton, *G. raimondii* and the model plant, *A. thaliana*, *T. cacao* and *O. sativa* were used to construct the phylogenetic tree. The proteins were aligned using Clustal W, a component of MEGA 6 software [52]. The phylogenetic tree was done using neighboring joint (NJ) method, as previously described by Magwanga et al. [43], in the construction of the phylogenetic tree for the cotton late embryogenesis abundant (LEA) proteins. The support for each node was tested with 1000 bootstrap replicates. In addition, we analyzed the gene structure of the cotton *ABC* genes while using an online tool gene structure displayer server (http://gsds.cbi.pku.edu.cn/).

### 2.9. Functional Characterization of the ABC Novel Gene through Virus Induced Gene Silencing (VIGS)

The Virus Induced Gene Silencing (VIGS) system was used to interfere with *Gh_D11G3432* gene expression in “Jinmian 20”, it is a type of upland cotton with high resistance to *V. dahlia* infection [17]. The cotton seeds were obtained from the Cotton Research Institute, Chinese Academy of Agricultural Science, Anyang, China (CRI-CAAS). The 477 bp fragment of the gene *Gh_D11G3432* was cloned with the specific primers Forward sequence: 5′GGAATTCAAGAAGAAATGGATATTTCTG3′; Reverse sequence: 5′CGGATCCCAACAAACTGAGAATAATTAC3′ and inserted into the vector pTRV2 of tobacco fragile virus (TRV) via EcoRI and BamHI digestion sites to construct a 35S promoter-driven pTRV2:Gh_D11G3432. The recombinant vector was transformed into the competent cells of *Agrobacterium tumefaciens* LBA4404 by the freeze-thaw method, and the bacterium LBA4404 containing pTRV1 vector was used as auxiliary bacteria. The cotyledons of cotton seedlings were infected by injection, and pTRV: PDS (Phytoene desaturase) was used as positive control. Plants without infection and empty vector pTRV: 00 were used as negative control, and, in each, three biological replications were done. The infected cotton seedlings were cultured in a 24 h darkness incubator under 16 h light/8 h dark photoperiod. After two weeks of treatment, the newly grown leaves exhibited albinism. Three days later, *V.* dahliae spore suspension (1.2 × 107 spores/mL) was inoculated. The control group was inoculated with the same volume of potato dextrose agarose (PDA) medium. The samples were taken 24 h after inoculation. *Ghactin7*, F: 5′ATCCTCCGTCTTGACCTTG3′; R: 5′TGTCCGTCAGGCAACTCAT3′, TRV1 primer, F: 5′TTACAGGTTATTTGGGCTAG3′; R: 5′CCGGGTTCAATTCCTTATC3′, TRV2 primer, F: 5′TGTTTGAGGGAAAAGTAGAGAACGT3′; R: 5′TTACCGATCAATCAAGATCAGTCGA3′. The Gh_D11G3432 a half RT-qPCR and RT-qPCR primer F: 5′CTCGACCTTGATACAA TCGA3′; R: 5′CAAACTGAGAATAATTACCC3′. The incidence of *V. dahliae* was investigated at 15 and 20 days after inoculation, and the plant disease index (DI) was calculated according to the formula: DI = 100 × ((n × the number of seedlings at level n))/(4 × the number of total seedlings), (n represents disease level (level 0, 1, 2, 3, 4). The incidence of DI was higher after inoculation. The higher that the DI was, the more serious the disease [17].

The mutant plants were further exposed to *V. dahliae* and morphological (leaf, stem, and roots tissues) and molecular evaluation carried out through RT-qPCR by the use of three known biotic stress responsive genes after 24 h periods the gene primer specifics are outlined in (Table 1), with *GhActin 7* as the reference gene.

## 3. Results

### 3.1. Different Response in the Three Cotton Species to V. dahliae Invasion

The effect of *V. dahliae* infection was severe in *G. raimondii*, as compared to the other two wild cotton species of the D genome, though it is worth noting that *G. thurberi* was highly resistant to *V. dahliae* when compared to *G. trilobum* (Figure 1A). The stem tissues were also examined after 25 dpi, the level of *V. dahliae* were monitored by the level of brown color change on the cortical and vascular bundle regions, the stem tissues that were obtained from *G. raimondii* showed higher disease infection when compared to the other two wild cotton species (Figure 1B). The root tissues were cut off tips, mounted onto the microscope slide, and then observed under florescence microscope. At 3 h of post inoculation, no infection thread was observed in *G. thurberi*, but a small degree of infection was noted in the roots of *G. trilobum* and *G. raimondii*, with an increase in exposure to the disease inoculant, the higher the infection level, in which at 6 h and 12 h of exposure, a significant level of infection was observed in *G. raimondii*, then *G. trilobum*, but a relatively lower level of disease infection was observed in the roots of *G. thurberi* (Figure 1C). The results elucidated that *G. thurberi* and *G. trilobum* were more resistant to *V. dahliae*, and they thus have the ability to resist infection for a relatively longer period of time when compared to *G. raimondii*.

### 3.2. Physiological and Biochemical Characteristics of the Three Cotton Species to V. dahliae Infection

The oxidant and the antioxidant enzymes were evaluated at varying time intervals after inoculation. There was a sharp increase in oxidant levels, hydrogen peroxide (H_2_O_2_), and malondialdehyde (MDA) in the leaves of *G. raimondii* as compared to the other two wild cotton species, *G. thurberi* and *G. trilobum* (Figure 2A,B). On the other hand, the concentration levels of the various antioxidant enzymes evaluated showed a higher concentration in the leaves of *G. thurberi* and *G. trilobum*, while there was significant reduction in the concentration in the leaves of *G. raimondii* (Figure 2C,E). When plants are exposed to any form of stress, the delicate balance between reactive oxygen production (ROS) and its elimination undergoes a drastic change, leading to overproduction and the excessive accumulation of ROS; high ROS concentration within the cells does cause massive cell destruction, which eventually lead to cell death [53]. The ability of the two wild cotton species, *G. trilobum* and *G. thurberi*, to induct more of the antioxidant enzymes is an indication that these plants could be harboring some significant genes that enable them to either resist or tolerate *V. dahliae*.

### 3.3. Sequencing Overview and Transcript Identification among the Three Cotton Species in Relation to V. dahliae Resistance

In order to determine the hidden mechanism underlying the resistance of the two wild cotton species to *V. dahliae* infection, a total of 81 cDNA samples were sequenced in three biological replications. The 81 cDNA samples were obtained from leaf, root, and stem tissues of the *G. thurberi*, *G. trilobum*, and *G. raimondii*. On average, a total of 397,970,060, 403,818,911.3, and 398,319,078.67 of clean reads representing 179.07 Gb, 181.72 Gb, and 179.25 Gb were obtained for *G. raimondii*, *G. thurberi*, and *G. trilobum*, respectively. The minimum values for the Q20 and Q30 percentages were 99.14% and 93.57%, respectively, as was observed in the sequence results for *G. thurberi*, the highest values for the Q20 and Q30 were 99.44% and 94.84%, respectively, and they were detected in the sequence analysis for the *G. raimondii*. The GC content ranged from 44.0–46.7% (Table 2). On average, 88.13% of reads were mapped to the reference genome, among which 52.15% were aligned to unique locations (Figure 1 and Appendix A). A total of 49,643, 53,247, and 49,283 novel transcripts were identified in *G. raimondii*, *G. thurberi*, and *G. trilobum*, and 35,682 genes were optimized with known structures.

### 3.4. Identification of the Differentially Expressed Genes (DEGs)

In the evaluation of the number of DEGs in various tissues of the three cotton species, a total of 1707, 2388, and 3434 DEGS were differentially expressed in the root tissues of *G. raimondii*, *G. thurberi* and *G. trilobum*, respectively. In relation to the stem and leave tissues, the DEGs in the stem for the three cotton species were, 2319, 3173, and 3789 for *G. raimondii*, *G. thurberi*, and *G. trilobum*, respectively, while in the leaves, 4305, 4830, and 3773 DEGs were detected in *G. raimondii*, *G. thurberi*, and *G. trilobum*, respectively (Figure 3A,B). The leave tissues showed a higher number of DEGs when compared to the roots and stem tissues, and indication that the leaves suffers more as a result of the *V. dahliae* infection, even though the first point of contact or the infection point being the root tissues. It may also mean that the plant’s defense mechanism is highly synchronized in the whole plant body [54]. Even though, more genes were found to be downregulated when compared to those which were upregulated, higher numbers of upregulated genes were detected for the two wild cotton species, *G. thurberi* and *G. trilobum*, an indication that the two wild cotton species harbored significant genes that are crucial for enhancing cotton tolerance to *V. dahliae* infection.

### 3.5. Gene Ontology (GO) and Kyoto Encyclopedia of Genes and Genomes (KEGG) Pathway Enrichment Analysis of the Differentially Expressed Genes

In the evaluation of the DEGs in terms of their predicted functions, all the DEGs were used to carry out GO and KEGG pathway analysis. GO analysis described the genes function into three categories, such as biological process (BP), cellular component (CC), and molecular function (MF). In each of the three cotton species, *G. raimondii*, *G. thurberi*, and *G. trilobum* DEGs were analyzed, and the analysis was done based on the various tissues profiled, the leaf, root, and the stem. Being the disease entry point is the root tissues, we critically looked into the various GO and KEGG pathways in which the DEGs at the root tissues were involved in amongst the three cotton species. All the three GO terms were assigned, at the biological process (BP) the following functions were detected oxidation-reduction process, protein phosphorylation, regulation of transcription-DNA-dependent, metabolic process, translation, signal transduction, vesicle mediated transport, among others. For cellular component (CC), functions, such as intracellular, membrane, nucleus, amongst others, and for the molecular functions (MF), functions such as protein binding, ATP binding, DNA binding, zinc ion binding, catalytic activity, nucleotide binding, amongst others (Figure 4A). It is interesting to note that, in all the three cotton species, MF, CC, and GO terms found were the same except for the biological functions in which some of the functions were detected in some species, but were not common among all the three, for instance, protein dephosphorylation, metal ion transport, nucleocytoplasmic transport, and transcription-DNA dependent were specific to *G. thurberi* DEGs at the root. The most frequent GO terms were “oxidation-reduction process”, “protein phosphorylation”, and “regulation of transcription-DNA dependent” among the BP, intercellular, membrane, and integral to membrane among the CC, while at the MF, GO functions, such as protein binding, ATP binding, and DNA binding were the most frequent. The corresponding genes of these significant terms, therefore, might play important roles in resistance to *V. dahliae* infection in cotton.

All the DEGs in the three cotton species, *G. raimondii*, *G. trilobum*, and *G. thurberi*, obtained from the root regions were largely enriched in various pathways, and grouped into level 1 and level 2, among the pathways within the level 1 were organismal systems, metabolism, genetic information processing, environmental information processing, and cellular processes, while those that were detected for level 2 were environmental adaptation, nucleotide metabolism, energy metabolism, membrane transport, transport and catabolism, replication, and repairs, among others (Figure 4B). Among all the KEGG pathways classification, the *ABC* genes were found to be dominant in environmental information processing (level 1) and membrane transport (level 2), with a total of 139 genes accounting for over 66% of all the genes involved in the membrane transport pathways. The detection of the *ABC* genes within the membrane transport and environmental information processing indicated that the proteins encoded by these genes have a vital role in enhancing the immunity of the plants, thus enabling the plants to resist *V. dahliae* infection. Moreover, GbSBT1 protein has been found to be located in plasma membrane and it has a positive role in enhancing plants immunity to *V. dahliae* infection [16]. The proteins encoded by the *ABC* genes were found to be abundantly sublocalized within the plasma membrane, which provides a further justification into their possible role in enhancing resistance to *V. dahliae* infection in cotton.

### 3.6. Identification, Phylogenetic and Sequence Analysis of the ABC Genes in the Diploid Cotton of the D Genome

A total of 166 proteins that were encoded by the *ABC* genes in *Gossypium raimondii* were identified. In the proteins encoded by the diploid cotton, *G. raimondii*, the *ABC* genes were highly varied in terms of their molecular weights (MW), protein lengths (aa), isoelectric points (pI), grand hydropathy average values (GRAVY), and even in their charge. The protein lengths ranged from 133–1890 aa, the molecular weights ranged from 14.76–210.467 kDa, their charge ranged from −10.5–45.5, while their GRAVY ranged from −0.62–0.361 (Appendix A), an indication that their GRAVY values were less than 1, several studies have shown that the stress responsive genes, their proteins exhibit relatively lower GRAVY values, more often less than 1, such as the Late embryogenesis abundant (LEA) proteins [55], cyclin dependent kinases (CDKs) [56], and dehydrins [57], thus the *ABC* genes could be having a function in enhancing disease resistance in cotton. All the ABC proteins were phylogenetically classified into three groups, namely clade 1, 2, and 3 (Appendix A), clade 1 and 3 were the largest, while clade 2 was the smallest. A novel *ABC* gene, *Gorai.007G244600* (ABC transporter F family member 5), homologous to *Gh_D11G3432* (ABC transporter F family member 5), and a member of the ABC proteins in clade 2, a tetraploid cotton gene, was identified for further functional analysis, (Figure 5A). In all the members of the subgroup 2 of clade 2, their gene structures were interrupted by introns, except Gorai.001G133500 and Gorai.006G124600 (Figure 5B), and harbored common motifs, such as motif 1, motif 2, and motif 5. The common motifs were further validated through protein alignment, and two unique motifs were found to be common, motif VPMVIISHDRAFLDQLCTKIVET and motif LDEPTNHLDIPSKEMLEEAI (Figure 5C). The proteins encoded by the *ABC* genes were found to be distributed in all the 13 chromosomes of *G. raimondii* with a single gene, Gorai.N013800 (Pleiotropic drug resistance protein 1) was found to be located within the scaffold regions, the highest gene density was detected in chromosome D507, with 21 accounting for 12.7% of all of the *ABC* genes in the diploid cotton, *G. remand*, though the least gene loci were detected on chromosomes D505, D510, and D513 with eight, five, and eight genes, respectively. In relation to subcellular localization prediction of the proteins encoded by the *ABC* genes, the highest proportions of the proteins were found to be embedded within the plasma membrane with 141 proteins, then the following cellular structures harbored ABC proteins in the proportions of 1–11, Golgi body, mitochondrion, nucleus, cytoplasm, and chloroplast, with one, two, two, nine, and 11 ABC proteins, respectively (Appendix A and Appendix A). The high proportions of the genes that were encoded by the *ABC* genes to be located within the plasma membrane, possibly showed that the proteins had a functional role, for instance, lysin motif-containing proteins 1, 7 and LysMe3 proteins have been predicted to be sub-localised within the plasma membrane, and their knock down resulted in impaired jasmonic acid, salicylic acid, and ROS generation, thus affected the activation of the defense genes and compromised mutant cotton resistance to *V. dahliae* [58]. Moreover, several plant plasma membrane-bound receptors, known as pattern recognition receptors, PRRs have been found to induce the first layer of innate immunity, which is also referred to as PAMP-triggered immunity (PTI) [59].

### 3.7. RT-qPCR Validation of the ABC Genes

In profiling the *ABC* genes, we selected a total of 44 genes out of the 160 genes that were detected in the RNA analysis of the ABC genes of *G. raimondii*. The genes were selected based on their phylogenetic tree grouping, RNA expression level (Appendix A), and the gene structure analysis. In carrying out RT-qPCR validation, root, stem, and leaf tissues were obtained from *G. raimondii*, *G. trilobum*, and *G. thurberi*. Higher expression levels were observed in the root tissues across the three cotton species, which indicated that more genes were inducted at the root level upon *V. dahliae* infection; this could be attributed to the fact that the roots are the primary entry point of the pathogens. The increased expression level of the *ABC* genes perhaps was to boost the immunity of the plant against *V. dahliae* infection. Moreover, significantly higher gene upregulation was observed in the various tissues of the wild cotton species, *G. thurberi* as compared to *G. raimondii* (Figure 6A–C). The result that was obtained is in agreement with previous findings, in which the more tolerant plants have been found to induct more genes when compared to less tolerant plants [55]. Furthermore, the wild cotton species are known to harbor significant and beneficial agronomic traits [43]. The expression nature of the *ABC* genes in the various tissues of cotton infected with *V. dahliae* is an indication that these groups of genes could be playing an important role in enhancing resistance to *V. dahliae* infection.

### 3.8. Virus Induced Gene Silencing (VIGS) Confirmation with the Gene Gh_D11G3432 (ABCF5) on Tetraploid Upland Cotton

The novel *ABC* gene used for further functional analysis of the *ABC* genes through virus induced gene silencing (VIGS) in cotton was identified based on the RNA expression and RT-qPCR validation. The upland cotton seedlings were infused with pTRV2:Gh_D11G3432, pTRV:PDS, pTRV:00, the none infused were used as wild type. After 12 days post inoculation, the cotton seedlings that were infused with the Agrobacterium carrying the TRV:PDS constructs exhibited white parches on the leaves (Albino type of appearance). The over 75% albino appearance occurred on the leaves after 20 days of post inoculation, which indicated that the gene knockdown was effectively done (Figure 7A,B). Furthermore, RT-qPCR assays were performed on leaves that were collected from the plants containing the TRV:00 empty vector and the TRV:Gh_D11G3432 (ABCF5)-construct. The transcript level of the target *Gh_D11G3432* (ABCF5)-gene decreased in the *Gh_D11G3432* (ABCF5)-silenced plants when compared with that in the TRV: 00 empty vector-infected (Figure 7C,D). Moreover, the disease index level was significantly higher in the mutant cotton seedlings as compared to the wild cotton (Figure 7E). These results collectively suggested that the target gene was successfully knocked down in cotton plants. In the evaluation of three known cotton disease responsive genes, cotton polyamine oxidase (GhPAO), cotton ribosomal protein L18 (GhRPL18), and cotton polygalacturonase-inhibiting protein-1 (GhPGIP1) on the *Gh_D11G3432* (ABCF5)-silenced plants, the positive controlled and the wild types under diseases treatment. All the three genes showed significantly low expression levels in the leaf tissues of the *Gh_D11G3432* (ABCF5)-silenced plants when compared with the wild and their controlled plants (Figure 7F). The down regulation of the three genes possibly indicated that the silenced gene had a functional effect in enhancing resistance to disease infection.

### 3.9. Evaluation of Oxidant and Antioxidant Enzymes Concentration Levels in the Leaf Tissues of the Gh_D11G3432 (ABCF5)-Silenced Plants under V. dahliae Infection

The concentration levels of the oxidants and antioxidant enzymes showed that the *Gh_D11G3432* (ABCF5)-silenced plants were highly infected when compared with the wild types and the controlled plants. The two oxidants evaluated were hydrogen peroxide (H_2_O_2_) and malondialdehyde (MDA); there was significant increased elevation of the two oxidants in the mutant cotton when compared to the wild types (Figure 8A,B). Similarly, the levels of the antioxidant enzymes were significantly low when compared to the wild types (Figure 8C–E).

## 4. Discussion

Cotton production globally is threatened by various biotic stress factors, the most common and with greater economic loss is the *Verticillium dahliae* infection [60]. *V. dahliae* infection, which is ranked top among the various forms of biotic stresses with higher economic losses in cotton production, because it causes *Verticillium* wilt, a terminal disease that results in plant death [7]. Controlling of the fungus that causes *Verticillium* wilt has posed a great challenge to plant breeders due to its wide host range, and its longevity in the soil in absence of suitable host [61]. The overall loss that is caused by *V. dahliae* infection in cotton production is estimated to be 30%, being almost equal to losses as a result of various abiotic stress factors, such as drought, salinity, and or cold stresses [43]. Moreover, hybrid sterility and intensive selection has eroded the genetic diversity of the elite cotton cultivars, thus narrowing the genetic diversity of the current cultivated forms of cotton genotypes [62]. The application of the wild cotton species have helped to broaden the genetic diversity and the introgression of some of the novel genes into the cultivated cotton genome [43].

Plants have been found to have evolved a number of strategies in order to tolerate the effects of abiotic and biotic stress factors, one of which is the induction of various stress responsive genes, such as the late embryogenesis abundant (*LEA*) genes [55], MYBs [63], cyclin dependent kinases (CDKs) [56], among others. In this research, we undertook to carry out deep sequencing analysis of three diploid cotton species, *G. raimondii*, *G. trilobum*, and *G. thurberi*, which were grown in highly infected rooting medium with *V. dahliae* fungus. The degree of disease tolerance among the three cotton species was varied; *G. thurberi* and *G. trilobum* were found to be more resistant when compared to *G. raimondii*. A total of 37,233 genes were obtained, among which the ATP-binding cassette (*ABC*) genes were highly mobilized and were found to be highly enriched in various pathways, such as ABC transporters (ko02010), Bile secretion (ko04976), and MicroRNAs (ko05206). The detection of miRNAs pathways to be associated with the *ABC* genes correlate positively to recent findings that suggest that efflux pumps of the ABC transporter family are dependent on the miRNA-mediated gene regulation mechanism. In addition, the ABC transporters are synchronized and the miRNA-guided network control the proteins that mediate the cell survival in changing environmental conditions [64,65]. Being the fungus block different transport channels within the plants; ABC proteins could perhaps be playing an important role in the regulation of the growth of the fungus within the various vascular bundles, such as phloem and the xylem, thus maintaining the translocation and transportation of photosynthates and water within the plant body [66]. Moreover, the various gene ontology annotations detected for the three GO terms, CC, MF, and BP, showed that the *ABC* genes could be playing a significant role in enhancing cotton resistance to *V. dahliae* infection.

Identification of the proteins encoded by the *ABC* genes in the diploid cotton of the D genome, *G. raimondii*, revealed that 166 proteins of the ABC protein domains are harbored in its genome, and the RNA deep sequencing revealed 160 proteins of the *ABC* gene domain, an indication that the sequencing of these genes in *G. raimondii* attained 96.4%, an indication that these proteins could be playing a role under *V. dahliae* infection in cotton, moreover 161 proteins that were encoded by the *ABC* genes were identified in the other two wild cotton species of the D genome, *G. trilobum* and *G. thurberi.* Phylogenetically, the proteins were grouped into three clades, with clade 1 and 2 containing the highest number of the ABC proteins; however, the novel gene was found to belong to members of clade two, and its ortholog to *Gorai.003G047600* (ABCF5) and *Thecc1EG001255*, the detection of an ortholog pair from *Theobroma cacao* showed that these genes evolved before the separation of the two plant species, being *T. cacao* and *Gossypium*, share a common ancestry. Across the phylogenetic tree, a number of ortholog pairs were detected, for instance, *Gorai.003G183800* and *Thecc1EG016555*; *Gorai.004G177600*; and, *Loc_Os11G34350,* among others. Due to the large number, only the novel genes with those members in clade 2 were analyzed for their structures, in which the majority of them harbored introns. The intron interruption is a burden to the genes, but the majority of the stress responsive genes have been found to harbor at least a single intron within their structures, for instance, the NAM, ATAF, and CUC (*NAC*) genes are known to be the top plant transcription factors in relation to stress tolerance, but for a number of them, their structures are permeated with introns [67].

RNA sequence and RT-qPCR analysis showed that higher numbers of the *ABC* genes were inducted in *G. thurberi* when compared to *G. trilobum* and *G. raimondii*, the results were in agreement with previous findings in which more stress tolerant genotypes have been found to induct more genes when compared to the sensitive ones, for instance, Magwanga et al. [43] found that more of the *LEA* genes were upregulated in the various tissues of G. tomentosum under drought stress when compared to *G. hirsutum* [43]. Moreover, more genes were found to be upregulated in the tissues of Verticillium wilt resistant hop cultivar as compared to the susceptible cultivar [68]. The gene expression results were further supported by the analysis of various oxidant and antioxidant enzymes on the leaf tissues of the three cotton species, MDA and H_2_O_2_ levels were higher in the leaves of *G. raimondii* and *G. trilobum*, but they were significantly reduced in the leaf tissues of *G. thurberi*, the opposite was true with the concentration levels of the various antioxidant enzymes evaluated. When plants are exposed to any form of stress, the delicate balance of reactive oxygen species (ROS) production and elimination become altered, leading to excessive production and the accumulation of ROS, which becomes intolerable to plant cell, leading to oxidative stress, and eventually sudden cell death, even though ROS are signal transducers [69].

Finally, we carried out functional analysis of the homeolog gene to *Gorai.007G244600* in upland cotton in order to understand whether the silencing of the gene could result into any changes within the plants under disease infection. Virus-induced gene silencing (VIGS) is an RNA silencing method that uses the recombinant viral vectors to transiently knock down the expression of the plant gene in a sequence-specific manner [70]. The mutant upland cotton seedlings exhibited a higher disease index, and the silenced gene was significantly downregulated on the leaf tissues. Moreover, evaluation of three plant biotic stress responsive genes, cotton polyamine oxidase (GhPAO), cotton ribosomal protein L18 (GhRPL18), and cotton polygalacturonase-inhibiting protein-1 (GhPGIP1) [71] were all downregulated in the leaves of the mutant but they were significantly upregulated in the wild and the positive controlled plants. The transgenic Arabidopsis plants with an overexpression of cotton polyamine oxidase (GhPAO) significantly improved their tolerance to *V. dahliae* and maintained putrescine [17]. In addition, overexpression of the ribosomal protein L18 (RPL18) gene in the model plant enhances resistance to *V. dahliae* [72]. The downregulation of these genes in the mutant cotton showed that the silenced gene had a significant role in boosting immunity of cotton to *V. dahliae* infection. Furthermore, the oxidant and antioxidant enzymes showed that the mutant cotton seedlings were highly affected by the disease infection as compared to the wild types.

## 5. Conclusions

Poor fiber quality and reduction in overall yield of cotton has been attributed to abiotic and biotic stress factors, and with *V. dahliae* infection accounting for 30% reduction in fiber quality and quantity [73]. The controlling of *V. dahliae* infection through conventional methods is not possible due to wide host range and the longevity of the infectious fungi in the soil, thus the application of molecular approach is the best mechanism to develop more resilient cotton genotypes with a higher resistance to *V. dahliae* infection. In this research work, we carried out RNA sequencing of three cotton species of the D genome, *G. raimondii*, *G. trilobum*, and *G. thurberi*, which were grown under high *V. dahliae* infested rooting medium. We obtained over 37,000 genes with varying levels of expression. Due to the huge number of genes obtained, we isolated the *ABC* gene family for further analysis based of KEGG pathways results of the top 20 highly expressed genes. From RNA sequencing, we obtained 160, 161, and 161 *ABC* genes in *G. raimondii*, *G. trilobum*, and *G. thurberi*, respectively. We further carried out whole genome identification of the *ABC* genes in *G. raimondii*, and found that it harbors a total of 166 *ABC* genes. Phylogenetically, the proteins encoded by the *ABC* genes were classified into three clades with varying physiochemical properties. A novel *ABC* gene, *Gorai.007G244600*, was identified, and it was found to be highly upregulated; its homolog in the tetraploid cotton was identified, and found to be *Gh_D11G3432*, which was used to carry out functional analysis of the *ABC* gene family under *V. dahliae* infection condition. The mutant cotton plants’ ability to resist and tolerate *V. dahliae* infection was significantly reduced, as evident by higher levels of oxidant enzymes and disease index scores. The results therefore revealed that *ABC* genes could be playing an important role in enhancing the resistance to *V. dahliae* infection.

## Figures and Tables

**Figure 1 genes-10-00110-f001:**
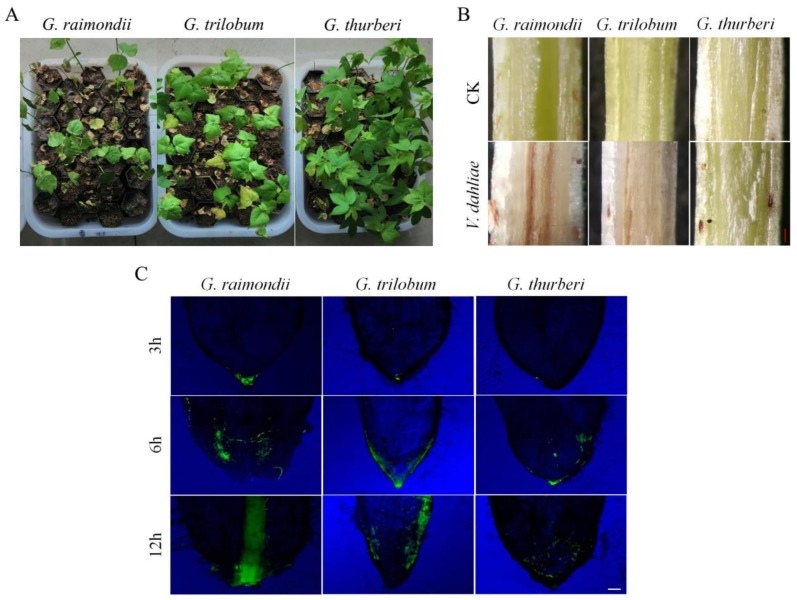
Physical and Florescence microscope examinations on stem and root samples of *G. raimondii*, *G. trilobum*, and *G. thurberi* infected with *V. dahliae*. (**A**) Disease symptoms at 25 dpi. (**B**) 25 dpi stem tissues (The lower part of the cotyledon node) showing brown color changes among three species on the cortical and vascular bundle regions (**C**) Root tips colonized by a green fluorescent protein (GFP) expressing isolate Vd-GFP at 3 h, 6 h, 12 h. CK: control. The scale bars approximately 10 µm.

**Figure 2 genes-10-00110-f002:**
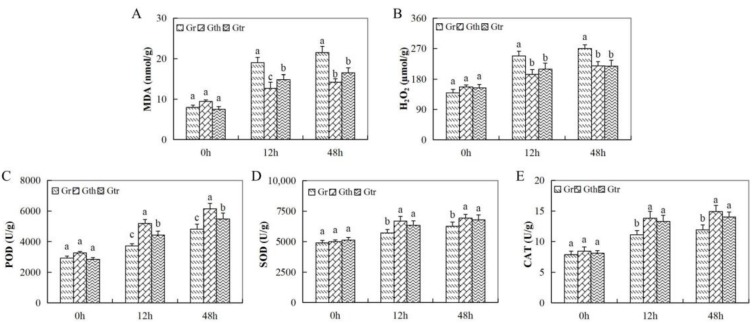
Oxidant and antioxidant enzymes concentration determination in the three cotton species under *V. dahliae* infection. (**A–E**): The changes of malondialdehyde (MDA), hydrogen peroxide (H2O2), peroxidase (POD), superoxide dismutase (SOD), and catalase (CAT) at different times post *V. dahliae* inoculation. Different letters indicate significant differences between the three cotton species (two-tailed; *p* <0.01).

**Figure 3 genes-10-00110-f003:**
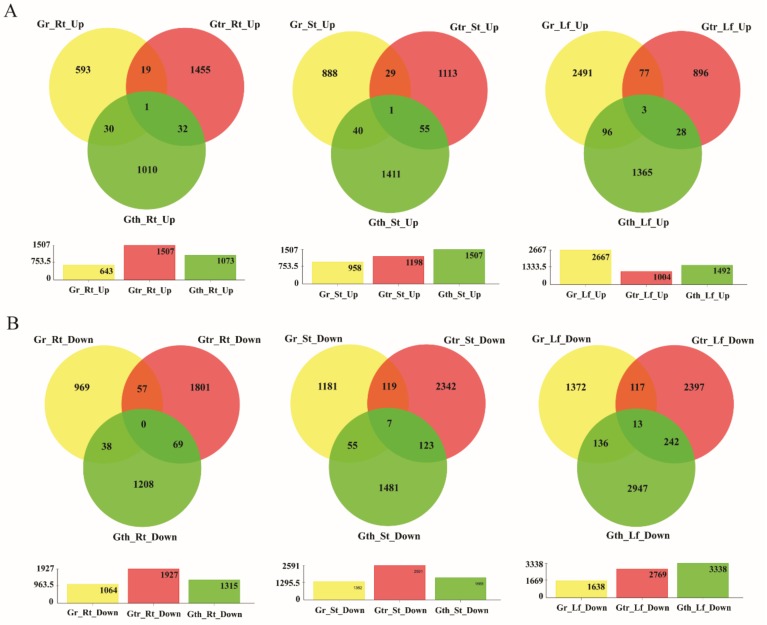
The Venn diagram of the differentially expressed genes (DEGs). The numbers indicate unique and common DEGs in three replicates for the different comparisons. Gr: *Gossypium raimondii*, Gtr: *Gossypium trilobum*, Gth: *Gossypium thurberi*, Rt: Root, St: Stem, and Lf: Leave. (**A**) Up regulation DEGs in three replicates for the different comparisons. (**B**) Down regulation DEGs in three replicates for the different comparisons. Y-axis: Number of DEGs.

**Figure 4 genes-10-00110-f004:**
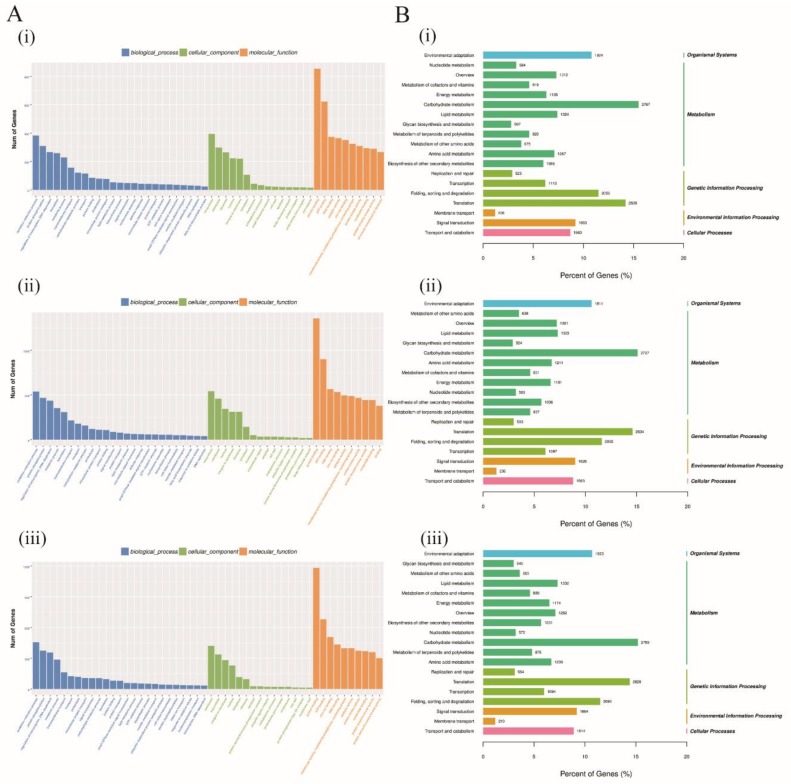
Gene Ontology (GO) and Kyoto Encyclopedia of Genes and Genomes (KEGG) pathway analysis of all the significant DEGs on the root tissues of *G. raimondii*, *G. trilobum*, and *G. thurberi*. (**A**) (i)–(iii) GO analysis of all the significant DEGs on the root tissues of *G. raimondii*, *G. trilobum*, and *G. thurberi.* (**B**) (i)–(iii) KEGG pathway analysis of all the significant DEGs on the root tissues of *G. raimondii*, *G. trilobum*, and *G. thurberi*.

**Figure 5 genes-10-00110-f005:**
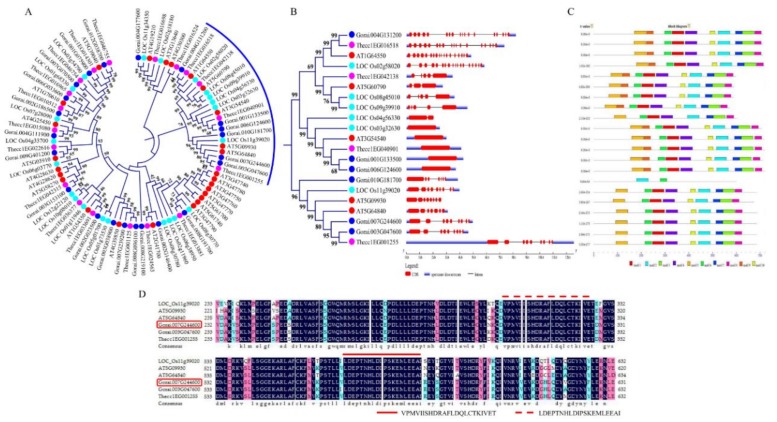
Phylogenetic tree, structural analysis and motif analysis of the Clade members of the cotton proteins encoded by the *ABC* genes in cotton. (**A**) Phylogenetic tree analysis of *G. raimondii* ABC proteins together with other plants. (**B**) Gene structures of the sub-region containing the novel gene used in RNAi analysis. (**C**) Motif identification through Multiple EM for Motif Elicitation (MEME) analysis. (**D**) Alignment of the sub region of the proteins containing the novel gene for RNAi.

**Figure 6 genes-10-00110-f006:**
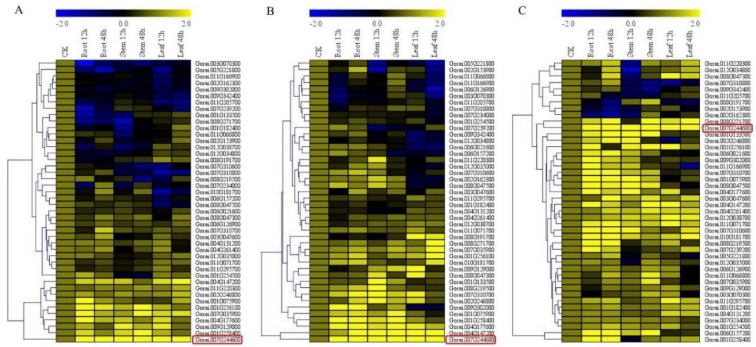
Quantitative Real-Time PCR (RT-qPCR) analysis of the expression of the *ABC* genes (**A**) RT-qPCR validation on the various tissues of *G. raimondii*. (**B**) RT-qPCR validation of the various tissues of *G. trilobum*. (**C**) RT-qPCR validation of the various tissues of *G. thurberi*. The heat map was visualized using Mev.exe program (Showed by log 10 values) in control and in treated 0 h, 12 h, and 48 h after disease exposure. Yellow–up regulated, blue–down regulated and black-no expression. CK: Control.

**Figure 7 genes-10-00110-f007:**
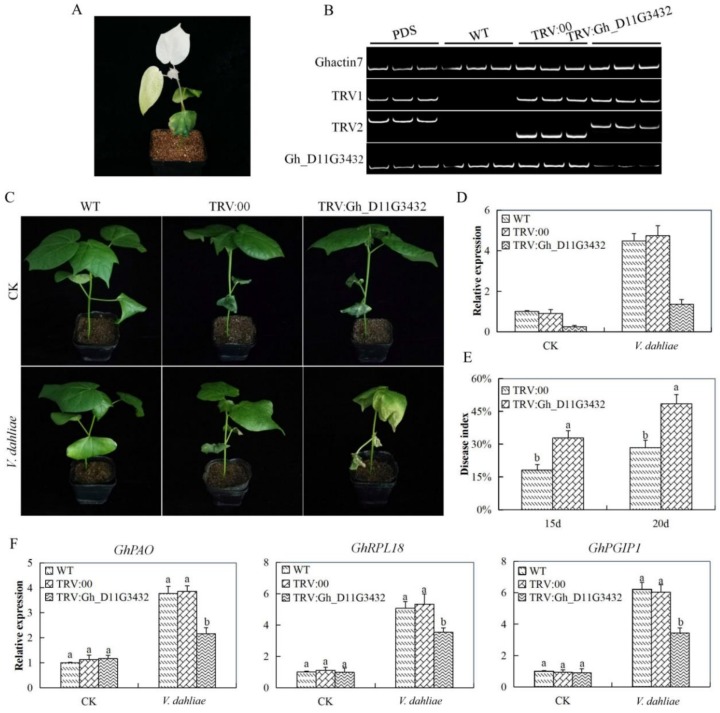
Phenotype observed in the silenced plants with the tobacco fragile virus (TRV): 00 empty vector, wild type plants, and *Gh_D11G3432* (ABCF5)-silenced plants at 20 days post inoculation. (**A**) Albino appearance on the leaves of the Phytoene desaturase (PDS) infused plants. (**B**) Gel electrophoresis in determining the effectiveness of gene silencing by the use of vector. (**C**) No obvious symptoms in the leaves of the TRV: 00-infected plant was observed. (**D**) RT-qPCR analysis of the change in the expression level of the *Gh_D11G3432* (ABCF5) gene in cotton plants treated with virus-induced gene silencing (VIGS). (**E**) Disease index on the VIGS and non VIGS plants under disease treated condition. (**F**) Stress responsive transcription analysis “TRV: 00” represents the plants carrying control the TRV2 empty vector; “TRV: *Gh_D11G3432*” represents the *Gh_D11G3432* (ABCF5)-silenced plants. Letters a/b indicate statistically significant differences (two-tailed, *p* <0.05). Error bars of the *Gh_D11G3432* (ABCF5) gene expression level represent the standard deviation of three biological replicates. CK: Control.

**Figure 8 genes-10-00110-f008:**
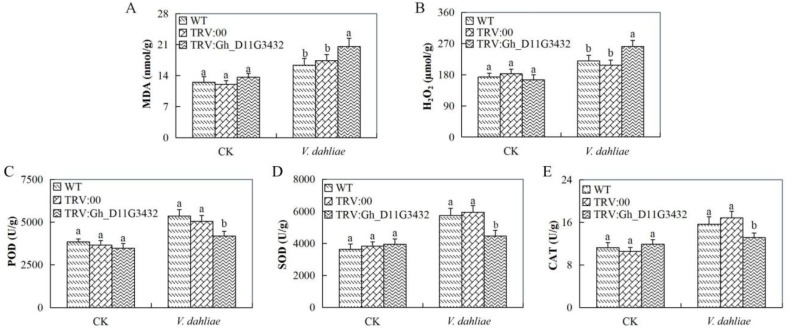
Determination of oxidants and antioxidant concentration levels in *Gh_D11G3432* (ABCF5). (**A**) Quantitative determination of MDA concentration. (**B**) Quantitative determination of H2O2 concentration. (**C**) Quantitative determination of POD concentration (**D**) Quantitative determination of SOD. (**E**) Quantitative determination of CAT in the leaves of wild-type and mutant cotton. In (**A–E**), each experiment was repeated three times. Bar indicates standard error (SE). Different letters indicate significant differences (ANOVA; *p* <0.05). CK: normal conditions.

**Table 1 genes-10-00110-t001:** Details of the biotic stress responsive genes.

Gene	Forward Sequence	Reverse Sequence
*GhPAO*	GGACAATCGCCTGAAACTGA	GAGGTCGCTTTGAAGAACACTA
*GhRPL18*	GCAGTTGAACAGATGTACGCT	ATTGCTTGGTGCTGTCCCTC
*GhPGIP1*	TCTGGTACAATCCCTGCCTC	CAGATCCAGCCTTGCCAAAC

**Table 2 genes-10-00110-t002:** Quality assessment of raw RNA-seq data.

Sample	Raw Data		Valid Data		Valid Ratio (Reads)	Q20%	Q30%	GC Content%
	Read	Base (G)	Read	Base (G)				
Gr_0L	45,671,298	6.85	45,174,949	6.77	98.89	99.42	95.62	46.67
Gr_12L	44,184,191	6.63	43,550,318	6.53	98.57	99.36	95.79	44.50
Gr_48L	41,258,547	6.19	40,872,607	6.13	99.07	99.45	95.43	44.67
Gr_0S	44,446,223	6.67	44,021,567	6.60	99.05	99.46	94.47	44.50
Gr_12S	46,330,216	6.95	45,813,186	6.87	98.88	99.50	94.66	44.17
Gr_48S	43,325,707	6.50	42,853,911	6.43	98.91	99.43	94.48	44.50
Gr_0R	46,091,584	6.91	45,704,989	6.86	99.16	99.50	94.28	44.33
Gr_12R	46,517,417	6.98	46,111,933	6.92	99.13	99.45	94.23	44.17
Gr_48R	44,322,955	6.65	43,866,601	6.58	98.98	99.47	94.56	44.00
Gth_0L	40,268,095	6.04	39,685,912	5.95	98.55	99.31	95.57	45.50
Gth_12L	46,036,569	6.91	45,436,039	6.82	98.68	99.28	95.50	45.00
Gth_48L	43,532,067	6.53	41,902,559	6.28	96.48	98.29	93.11	45.83
Gth_0S	43,844,243	6.58	43,353,597	6.50	98.88	99.36	94.00	45.00
Gth_12S	47,615,319	7.14	46,946,61	7.04	98.60	99.17	92.61	44.50
Gth_48S	43,248,223	6.49	42,695,944	6.40	98.72	99.27	93.47	44.67
Gth_0R	44,767,686	6.71	44,295,005	6.65	98.95	99.16	91.83	44.50
Gth_12R	52,161,200	7.82	51,588,587	7.74	98.91	99.15	92.61	44.17
Gth_48R	48,449,214	7.27	47,914,659	7.19	98.90	99.28	93.45	44.33
Gtr_0L	44,308,137	6.65	43,769,161	6.56	98.78	99.36	95.57	45.50
Gtr_12L	42,008,271	6.30	41,421,274	6.21	98.59	99.23	94.87	45.50
Gtr_48L	46,357,695	6.96	45,889,514	6.88	98.99	99.47	95.55	46.17
Gtr_0S	44,317,347	6.65	43,840,067	6.58	98.92	99.29	92.52	44.50
Gtr_12S	44,015,373	6.60	43,571,182	6.54	98.99	99.30	92.83	44.50
Gtr_48S	49,119,885	7.37	48,625,743	7.29	98.99	99.24	92.62	45.33
Gtr_0R	46,559,101	6.99	46,138,478	6.92	99.10	99.33	92.80	44.83
Gtr_12R	43,146,409	6.47	42,689,842	6.40	98.95	99.47	94.06	44.00
Gtr_48R	42,801,330	6.42	42,373,817	6.36	98.99	99.31	92.90	44.17

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
