# Peer review of "RNA-Sequencing, Physiological and RNAi Analyses Provide Insights into the Response Mechanism of the ABC-Mediated Resistance to Verticillium dahliae Infection in Cotton"

_genes, 2019, doi:10.3390/genes10020110_

Round 1
Reviewer 1 Report
In this manuscript, Dong et al analyze the gene expression changes induced by Verticillium dahliae in cotton. To this end they compared the gene expression pattern between two wild accessions that are relatively resistant to the fungi infection. This analysis lead the authors to identify several ABC transporters as candidate genes induced by the infection and potentially needed for the progression of the fungi. The authors further characterize one of these ABC transporter through VIGS, showing that artificial downregulation of an ABC transporter decreases the resistance of cotton to Verticillium infection. The amount of work included in this manuscript is substantial and of very good quality. The manuscript is moreover well written and the data is correctly analyzed. I only have a couple of suggestions that will help to clarify some aspects of the manuscript:
1) Although it is explained in the materials and methods section I think it would be worth it to include a description of where the 81 cDNA libraries come from (what tissues, how many replicates per tissue) in section 3.3.
2) Section 3.8 needs an introduction explaining how was the experiment designed, why the gene selected etc.
3) Also it is needed a paragraph explaining how many bioreplicates and technical replications where assayed in the VIGS experiment.
4) The discussion needs a better introductory paragraph. The one present at this stage is very confusing and needs substantial grammatical revision.
5) Typos and minor language corrections:
-Line 96: "A number of researches have been done.." Does not sound correct to me.
- Line 343: "Differntially" Typo, please change
-Line 402: "was found to homologous to .." Does not sound correct to me.
-Line 404: "All the member of the subgroup 2 of clade 2, ..." Maybe it will sound better adding "in" at the begining of the sentence.
-Line 464: there is a "d" that might be connected to another word.
-Lines 464 to 469: Please rewrite these two sentences and find a connector between them or divide them in a bigger number of sentences.
Author Response
REVIEWER 1
Open Review
(x) I would not like to sign my review report
( ) I would like to sign my review report
English language and style
( ) Extensive editing of English language and style required
( ) Moderate English changes required
(x) English language and style are fine/minor spell check required
( ) I don't feel qualified to judge about the English language and style
Response: Thanks so much, we have carried out thorough review, and all language errors have been corrected.
Yes | Can be improved | Must be improved | Not applicable | |
Does the introduction provide sufficient background and include all relevant references? | (x) | ( ) | ( ) | ( ) |
Is the research design appropriate? | (x) | ( ) | ( ) | ( ) |
Are the methods adequately described? | (x) | ( ) | ( ) | ( ) |
Are the results clearly presented? | (x) | ( ) | ( ) | ( ) |
Are the conclusions supported by the results? | (x) | ( ) | ( ) | ( ) |
Comments and Suggestions for Authors
In this manuscript, Dong et al analyze the gene expression changes induced by Verticillium dahliae in cotton. To this end they compared the gene expression pattern between two wild accessions that are relatively resistant to the fungi infection. This analysis lead the authors to identify several ABC transporters as candidate genes induced by the infection and potentially needed for the progression of the fungi. The authors further characterize one of these ABC transporter through VIGS, showing that artificial downregulation of an ABC transporter decreases the resistance of cotton to Verticillium infection. The amount of work included in this manuscript is substantial and of very good quality. The manuscript is moreover well written and the data is correctly analyzed. I only have a couple of suggestions that will help to clarify some aspects of the manuscript:
1) Although it is explained in the materials and methods section I think it would be worth it to include a description of where the 81 cDNA libraries come from (what tissues, how many replicates per tissue) in section 3.3.
Response: Thanks so much, we added the information as advised
2) Section 3.8 needs an introduction explaining how was the experiment designed, why the gene selected etc.
Response: Thanks so much, information added “ the novel gene used for further functional analysis, was selected based on the RNA sequencing and RT-qPCR validation
3) Also it is needed a paragraph explaining how many bioreplicates and technical replications where assayed in the VIGS experiment.
Response: In each set up, three replication was done.
4) The discussion needs a better introductory paragraph. The one present at this stage is very confusing and needs substantial grammatical revision.
Response: Adjustment
5) Typos and minor language corrections:
-Line 96: "A number of researches have been done.." Does not sound correct to me.
- Line 343: "Differntially" Typo, please change
Response: corrected
-Line 402: "was found to homologous to .." Does not sound correct to me.
Response: the statement corrected
-Line 404: "All the member of the subgroup 2 of clade 2, ..." Maybe it will sound better adding "in" at the begining of the sentence.
Response: Added
-Line 464: there is a "d" that might be connected to another word.
Response: thanks, “d” was part of the previous word, the space deleted.
-Lines 464 to 469: Please rewrite these two sentences and find a connector between them or divide them in a bigger number of sentences.
Response: The statement corrected.
Reviewer 2 Report
This is a review of manuscript entitled "RNA-Sequencing, Physiological and RNAi Analyses Provide Insights into the Response Mechanism of the ABC-Mediated Resistance to Verticillium dahlia Infection In cotton" submitted to “Genes”. In this research article, the authors used RNA sequencing to study three wild cotton species infected with Verticillium dahlia and found that Gossypium trilobum and Gossypium thurberi were more resistant to V. dahliae infection than Gossypium raimondii.
Well designed and executed experiments to show that ATP -binding cassette (ABC) proteins that are crucial to defend against V. dahliae infection in cotton. Below are some minor comments for the authors to consider for improving the clarity of this manuscript.
Minor comments:
1. Abstract. Correct sentence as “Verticillium wilt caused by Verticillium dahliae, results in massive annual yield losses and fiber quality decline in cotton”.
2. Introduction. Correct sentence as “The fungus get its entry point into the plants
through the roots”.
3. Introduction. This sentence is not clear. “It not only lays a foundation for the study of the transport and accumulation of cotton metabolites, but also provides a reference for the study on the mechanism of cotton autoimmune fungi”.
4. Introduction. Add one or two sentences about cotton strains Gossypium trilobum and Gossypium thurberi used in this study.
5. Materials and methods, 2.3. Mention the full form of SPSS.
6. Materials and methods, 2.4. Change the phrase to “biological replicates”.
7. Results. Remove ‘was’ from this sentence. “the stem tissues obtained from G. raimondii was showed higher disease infection”.
8. Results, 3.2. Correct heading as “Physiological and Biochemical Characteristics of the Three Cotton Species to V. dahliae Infection”.
9. Results, 3.2. Correct as “high ROS concentration within the cells does cause massive cell destruction, which eventually leads to cell death”.
10. Figure-1, 6, 7. Mention in legend what CK stands for.
11. Figure-2. Mention in legend what a and b stand for.
12. Figure-3. Mention y-axis label for bar graphs.
13. Results, 3.5. Correct as “we critically looked into the various GO and KEGG pathways”.
14. How did the GO and KEGG pathways look for the 3 cotton species in leaves and stems? Discuss about this in discussion section.
15. Result, 3.6. This sentence is not clear. “But being a single novel ABC gene was identified for further functional analysis, the gene Gorai.007G244600 (ABC transporter F family member 5) which was found to homologous to a tetraploid cotton gene”.
16. Figure-5. In the legend mention full form of MEME.
17. Figure-7. Mention full form of PDS.
18. Discussion. Correct sentence as “V. dahliae infection, ranked among the various forms of biotic stresses with higher economic losses in cotton production, because it causes Verticillium wilt, a terminal disease which results in plant death”.
19. Discussion. Correct as “because it results in up to 30% yield reductions”.
20. Discuss in what miRNA pathways are ABC genes enriched and their relevance to resistance against V. dahliae infection.
21. Discussion. Mention full form of NAC.
Author Response
REVIEWER 2
Open Review
(x) I would not like to sign my review report
( ) I would like to sign my review report
English language and style
( ) Extensive editing of English language and style required
(x) Moderate English changes required
( ) English language and style are fine/minor spell check required
( ) I don't feel qualified to judge about the English language and style
Response: Thanks so much, we have carried out thorough review, and all language errors have been corrected.
Yes | Can be improved | Must be improved | Not applicable | |
Does the introduction provide sufficient background and include all relevant references? | (x) | ( ) | ( ) | ( ) |
Is the research design appropriate? | (x) | ( ) | ( ) | ( ) |
Are the methods adequately described? | (x) | ( ) | ( ) | ( ) |
Are the results clearly presented? | (x) | ( ) | ( ) | ( ) |
Are the conclusions supported by the results? | (x) | ( ) | ( ) | ( ) |
Comments and Suggestions for Authors
This is a review of manuscript entitled "RNA-Sequencing, Physiological and RNAi Analyses Provide Insights into the Response Mechanism of the ABC-Mediated Resistance to Verticillium dahlia Infection In cotton" submitted to “Genes”. In this research article, the authors used RNA sequencing to study three wild cotton species infected with Verticillium dahlia and found thatGossypium trilobum and Gossypium thurberi were more resistant to V. dahliae infection thanGossypium raimondii.
Well designed and executed experiments to show that ATP -binding cassette (ABC) proteins that are crucial to defend against V. dahliae infection in cotton. Below are some minor comments for the authors to consider for improving the clarity of this manuscript.
Minor comments:
1. Abstract. Correct sentence as “Verticillium wilt caused by Verticillium dahliae, results inmassive annual yield losses and fiber quality decline in cotton”.
Response: corrected
2. Introduction. Correct sentence as “The fungus get its entry point into the plants
through the roots”.
Response: corrected
3. Introduction. This sentence is not clear. “It not only lays a foundation for the study of the transport and accumulation of cotton metabolites, but also provides a reference for the study on the mechanism of cotton autoimmune fungi”.
Response: the statement corrected
4. Introduction. Add one or two sentences about cotton strains Gossypium trilobum andGossypium thurberi used in this study.
Response: information added
5. Materials and methods, 2.3. Mention the full form of SPSS.
Response: Full meaning added
6. Materials and methods, 2.4. Change the phrase to “biological replicates”.
Response: Changed
7. Results. Remove ‘was’ from this sentence. “the stem tissues obtained from G. raimondii was showed higher disease infection”.
Response: removed
8. Results, 3.2. Correct heading as “Physiological and Biochemical Characteristics of the ThreeCotton Species to V. dahliae Infection”.
Response: corrected
9. Results, 3.2. Correct as “high ROS concentration within the cells does cause massive cell destruction, which eventually leads to cell death”.
Response: corrected
10. Figure-1, 6, 7. Mention in legend what CK stands for.
Response: CK: control
11. Figure-2. Mention in legend what a and b stand for.
Response: the meaning added within the figure legend
12. Figure-3. Mention y-axis label for bar graphs.
Response: the Y- axis specified within the figure legend
13. Results, 3.5. Correct as “we critically looked into the various GO and KEGG pathways”.
Response: corrected
14. How did the GO and KEGG pathways look for the 3 cotton species in leaves and stems? Discuss about this in discussion section.
Response: information on KEGG and GO added in the discussion part
15. Result, 3.6. This sentence is not clear. “But being a single novel ABC gene was identified for further functional analysis, the gene Gorai.007G244600 (ABC transporter F family member 5) which was found to homologous to a tetraploid cotton gene”.
Response: The incoherent nature of the sentence corrected
16. Figure-5. In the legend mention full form of MEME.
Response: Added
17. Figure-7. Mention full form of PDS.
Response: Added
18. Discussion. Correct sentence as “V. dahliae infection, ranked among the various forms of biotic stresses with higher economic losses in cotton production, because it causes Verticillium wilt, a terminal disease which results in plant death”.
Response: Corrected
19. Discussion. Correct as “because it results in up to 30% yield reductions”.
Response: corrected
20. Discuss in what miRNA pathways are ABC genes enriched and their relevance to resistance against V. dahliae infection.
Response: The detection of miRNAs pathways to be associated with the ABC genes, correlate positively to recent findings which suggest that efflux pumps of the ABC transporter family are dependent on the miRNA-mediated gene regulation mechanism. In addition, the ABC transporters are synchronised and the miRNA-guided network control the proteins that mediate the cell survival in changing environmental conditions (Pang et al. 2013; Geisler et al. 2017)
Geisler M, Aryal B, Di Donato M, Hao P (2017) A critical view on ABC transporters and their interacting partners in auxin transport. Plant Cell Physiol 58:1601–1604 . doi: 10.1093/pcp/pcx104
Pang K, Li Y, Liu M, et al (2013) Inventory and general analysis of the ATP-binding cassette (ABC) gene superfamily in maize (Zea mays L.). Gene 526:411–428 . doi: 10.1016/j.gene.2013.05.051
Response: added